# Subcutaneous Emphysema of the Orbit after Nose-Blowing

**Riku Mihara** [1]🆔 **, Yasuo Murai** [1,2,]*🆔 **, Shun Sato** [1,2]🆔 **, Fumihiro Matano** [1,2] **and Akio Morita** [1]

1   Department of Neurological Surgery, Nippon Medical School, 1-1-5 Sendagi, Bunkyo-ku,
    Tokyo 113-8602, Japan; s13-095mr@nms.ac.jp (R.M.); s3049@nms.ac.jp (S.S.); s00-078@nms.ac.jp (F.M.);
    amor-tky@umin.ac.jp (A.M.)
2   Department of Neurosurgery, Hakujikai General Hospital, 5-11-1 Shikahama, Adachi-ku,
    Tokyo 123-0864, Japan
*   Correspondence: ymurai@nms.ac.jp; Tel.: +81-3-3822-2131

**Abstract:** Orbital emphysema after nose-blowing is an uncommon condition and can appear without a trigger. Herein, we reported a case of orbital emphysema after nose-blowing and performed a literature review. A 68-year-old man fell and sustained an injury near his left orbit. No symptoms were noted. He noticed a left periorbital swelling after blowing his nose. Through computed tomography examination, he was diagnosed with subcutaneous emphysema. There are no previous reports that have reviewed the clinical features, need for surgery, and severity of symptoms of subcutaneous emphysema after nasal swallowing due to different factors. We retrospectively analyzed a cohort of 48 cases by searching PubMed to clarify these issues. Regarding the emphysema trigger, 21 cases had an injury or had previously undergone surgery. In 34 cases, conservative treatment was required, while surgery was selected in the acute phase in 6 cases and after the acute phase as a radical cure in 8 cases. Reduced visual acuity, diplopia, exophthalmos, facial hypoesthesia, and color disorders were noted and were more common among surgical cases. The literature review revealed no association between fracture location and the need for surgery; furthermore, surgery was less required in non-trauma cases, excluding osteoma, than in trauma cases ($p = 0.0169$). Our study reveals that a strict follow-up examination of visual symptoms is necessary for the first 2 days in cases of subcutaneous emphysema caused by nose blowing after facial trauma.

**Keywords:** emphysema; fracture; nose-blowing; orbit; trauma

## 1. Introduction

Orbital emphysema is a condition wherein air is stored in the orbital or eyelid soft tissue. Clinically, it is relatively rare. Regarding the etiology, irrespective of the presence of head or facial injury, there have been previous reports of orbital emphysema due to nose-blowing [1–32]. Surgery is rarely required [3,8,11,12,15–17,20,22,26,33] as the condition of a significant proportion of patients is improved with conservative treatment, which includes avoiding nose-blowing after the correct diagnosis is established [1,2,4,5,7,9,10,13,14,18–21,23–32]. It is possible that the increase in orbital pressure can lead to exophthalmos, impaired eye movement, and reduced visual acuity. In cases of orbital emphysema with visual disorders, there have also been reports of patients who required acute surgical decompression, such as needle aspiration [3,8,15,20,22,26,33]. Due to the rarity of orbital emphysema, its causes, symptoms, and treatment are controversial. We encountered a case of orbital emphysema caused by nose-blowing after head trauma. A search of previous literature works revealed no reviews that have summarized age, sex, progression of trauma or surgery, fracture location, treatment, other symptoms, outcomes, or the effect of antibiotics. Furthermore, no study has statistically analyzed the trends of cases requiring surgery, which was performed for the first time in this study. Diseases that can trigger orbital emphysema due to nose-blowing include facial trauma, medical treatment, and osteoma, but we did not identify any reports that compared the symptoms of these triggers, their differences in severity, and

the need for surgical treatment. The purpose of this study was to examine these issues and to contribute to the formulation of treatment policies.

## 2. Case Description

A 68-year-old man fell and sustained an injury near his left orbit 3 days before presenting to our institution. Initially, no swelling was noted; however, a swelling later appeared around the left eye. No fever, epistaxis, redness, or disorders of eye movement were noted after the head injury, and the patient's medical history was unremarkable. Although the swelling appeared 18 h before the patient presented to our hospital, no reduction in the swelling was noted. The patient reported hearing a "swoosh" sound after blowing his nose, followed by the appearance of swelling around the left eye. Although there was no pain on palpation, snowball crepitation was detected. Facial computed tomography examination showed subcutaneous emphysema around the orbit and a medial orbital wall fracture that continued to the ethmoid sinus (Figure 1A,B). He was diagnosed with subcutaneous emphysema due to nose-blowing. As no decrease in the swelling was observed, the patient was instructed to avoid blowing his nose. The swelling disappeared after a 10-day follow-up.

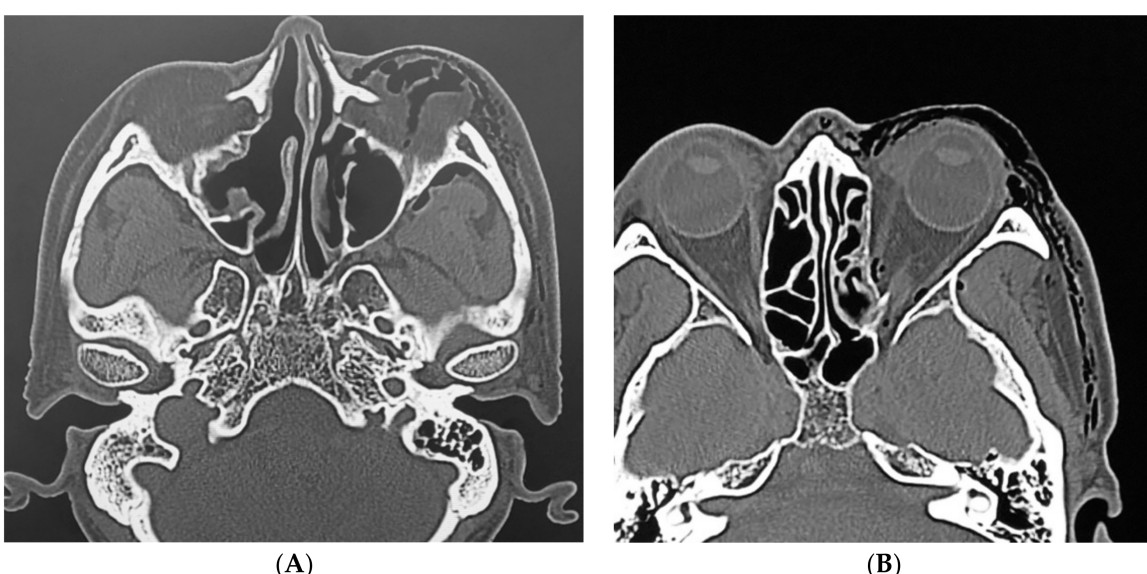

(**A**)    (**B**)

**Figure 1.** (**A**) Plain head computed tomography scan. A subcutaneous hematoma around the left orbit and temporal muscle is shown. The subcutaneous emphysema is visible in the orbit, which is continuous with the ethmoid sinus. (**B**) The left medial orbital wall is fractured.

### 2.1. Review

We reviewed previous articles that reported the clinical characteristics and treatment of patients with orbital emphysema. A systematic review of the literature was conducted in accordance with the PRISMA guidelines (Figure 2).

We searched the PubMed database for cases diagnosed with orbital emphysema caused by nose-blowing. Our key search terms were "nose blowing emphysema" or "nose-blowing, emphysema." We performed a search for "any field," from which 69 studies were retrieved. We also searched on Google using keywords from which we retrieved six references. We also thoroughly researched the literature from references listed in Pubmed and Google online search hits. The initial search was completed at the end of September 2020. Among these 75 references, six references were from conference abstracts. Six non-English language articles were also removed from this review. Ten papers that did not cover the location of subcutaneous emphysema were also removed. Four papers that did not explicitly state that nose-blowing was the cause were excluded. One report that did not provide detailed case information was excluded. Finally, data from 48 cases (Tables 1 and 2)

were reviewed. From these studies, we noted that the authors investigated the age, sex, trigger, time from exposure to trigger treatment, other symptoms, and outcomes of orbital emphysema. However, none of the 48 studies summarized age, sex, proceeding trauma or surgery, fracture location, treatment, other symptoms, outcomes, and effect of antibiotics; thus, this is the first study to mention the aforementioned variables.

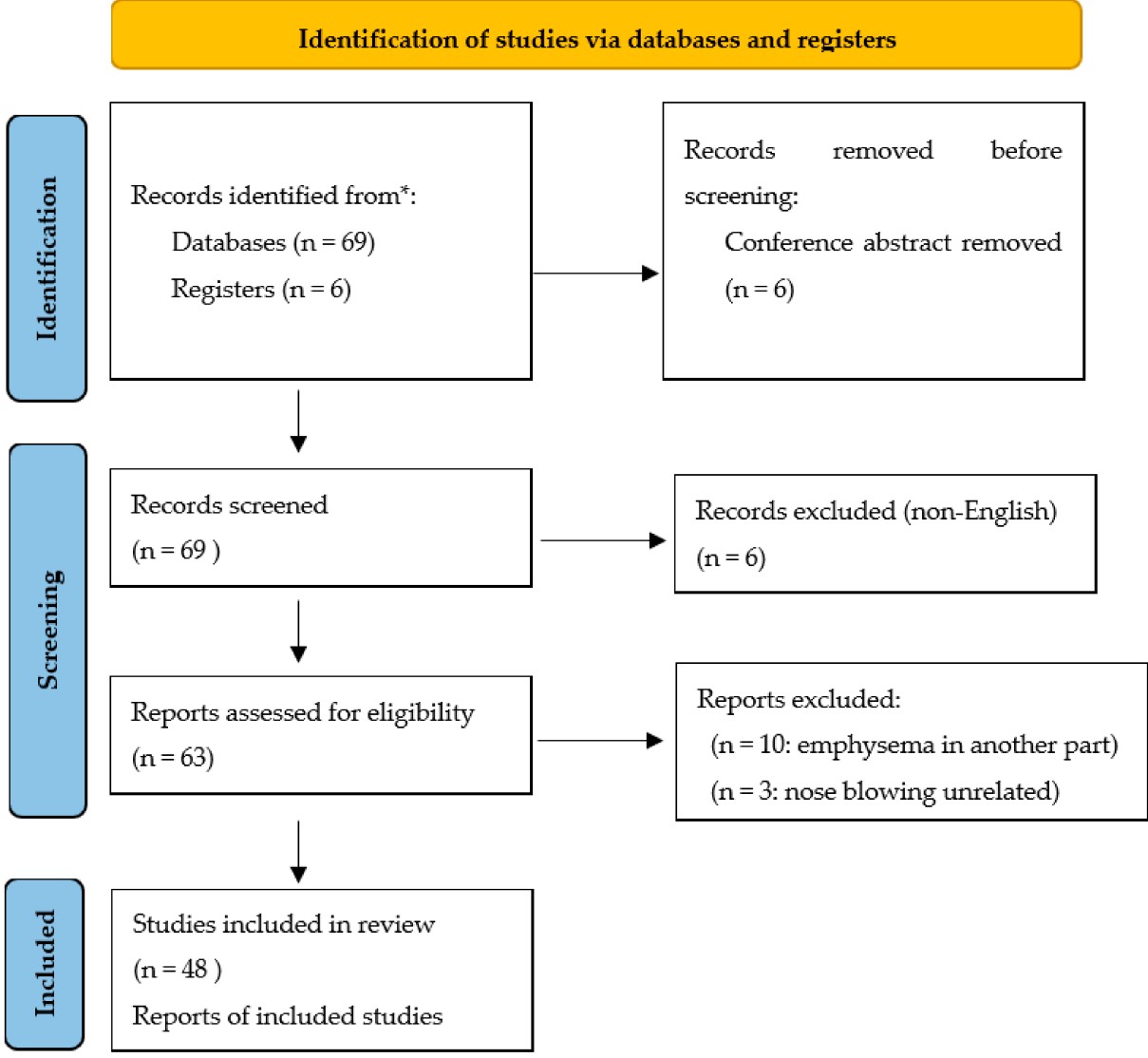

**Figure 2.** PRISMA flow diagram. *: Pubmed and Google.

**Table 1.** Characteristics of reported patients with orbital emphysema due to nose-blowing.

| Cases | |
|---|---|
| Male: Female | 31:17 |
| Age (years) | |
| Lange/Mean/Median | 11–80/40.7/38 |
| Treatment | |
| Conservative | 34 |
| Surgical | 14 (within 48 h: six; after acute phase: eight) |
| Trauma or other trigger | |
| None | 26 |
| Facial trauma | 18 |
| Surgery | 4 |

**Table 2.** Comparison between surgical and conservative cases.

| | Surgical | Conservative | *p*-Value |
|---|---|---|---|
| Number (Male) | 14 (9) | 34 (22) | 0.614 |
| Age: Range/Mean/Median | 16–68/40.6/39 | 11–80/57.4/36 | 0.65 |
| Trauma (+):(-) | 9:5 | 13:12 | 0.0921 |
| Without osteoma | 9:2 | 13:11 | *0.0169* |
| Symptoms | | | |
| Visual acuity deficit | 6 | 5 | |
| Limited eye movement or diplopia | 5 | 5 | |
| Exophthalmos | 8 | 1 | |
| Hypo-aesthesia | 2 | 3 | |
| Visual field defect | 1 | 1 | |
| Color disorder | 0 | 2 | |
| Location of fracture | | | |
| Total | 12 | 29 | 0.107 |
| Ethmoidal bone | 5 | 17 | |
| Maxillary bone | 5 | 9 | |
| Zygoma | 2 | 3 | |

*2.2. Statistical Analyses*

Statistical significance was determined using Fisher's exact test due to heterogeneous and small sample sizes. Non-normally distributed continuous variables were compared between groups using the Mann–Whitney U test. In all analyses, *p*-values of <0.05 were considered statistically significant, and a commercially available software package (JMP Pro 13, SAS Institute, Cary, NC, USA) was used for all analyses.

*2.3. Literature Review Results*

We studied 17 cases wherein a period between the occurrence of facial trauma and the development of emphysema was described. In nine, one, one, and one cases, facial trauma occurred within <24 h, <10 days, after 5 months, and after 10 years, respectively. There were more male than female patients (31 male cases; 65%). The medial orbital wall (22 cases; 46%) was the most common site of the fracture, and the inferior wall (14 cases;

29%) was the second most common site. There were seven cases (15%) wherein no fracture could be observed; however, there were four osteoma cases (8%). Regarding the triggers for emphysema, less than half of the cases had head injuries or a history of head and neck surgery (21 cases; 44%), while 18 and four cases (38%, 8%) were associated with trauma and surgery, respectively (in one case, both were noted). In 34 cases (71%), only conservative treatment was required. In six cases (13%), surgery was required in the acute phase (within 48 h from the first symptom occurrence), while in eight cases (16%), surgery was selected after the acute phase as radical treatment. In seven of these cases, needle aspiration or incision for decompression was selected (four cases of non-decompression surgery for fracture and three cases of surgery for osteomas).

Reduced visual acuity, limitation of eye movement or diplopia, exophthalmos, facial hypoesthesia, visual field loss, and color disorders (in 11, 10, nine, five, two, and two cases; 23%, 21%, 19%, 10%, 4%, and 4%, respectively) were reported as the main symptoms of clinical orbital emphysema. Visual dysfunction occurred simultaneously with nose blowing; however, in one case, visual dysfunction occurred 48 h later. In the surgical cases, reduced visual acuity, eye movement limitation or diplopia, exophthalmos, facial hypoesthesia, and visual field loss (6, 5, 8, 2, and 1 cases; 43%, 36%, 57%, 14%, and 7%, respectively) were experienced. In all cases, the symptoms were improved. No recurrence of orbital emphysema was reported after 1 year. Among the patients who underwent conservative treatment, antibiotics were used in 24 of 34 cases (71%), with surgical procedures performed in 11 of 14 cases (76%). Notably, we recognized that no patient had an infection.

A statistical study of surgical and nonsurgical cases was performed to identify factors that may lead to the need for surgical treatment (Table 2). Surgical treatment was performed in 29.3% and 29.4% of male and female patients, respectively, with no statistically significant difference observed in sex ($p = 0.614$). Surgical treatment was required in 40.9% and 19.2% of trauma and non-trauma cases, respectively, with no statistically significant difference ($p = 0.0921$), and this was examined in more detail. Non-trauma cases included four cases of osteoma, which required surgery. Therefore, when cases of osteomas were excluded from the non-trauma cases, only 9.1% of the non-trauma cases underwent surgery. Non-trauma cases, excluding osteoma, were less likely to undergo surgery than trauma cases ($p = 0.0169$). Finally, the need for surgery was examined according to the presence or absence of a fracture. Surgery was performed in 30% and 0% of patients with and without fractures, respectively, and no statistically significant difference was observed in the need for surgery according to the presence or absence of a fracture ($p = 0.107$).

## 3. Discussion

We encountered a patient with orbital emphysema caused by nose-blowing after sustaining head trauma. This case was conservatively managed because the symptoms were minimal. The subcutaneous emphysema resolved without performing surgery and has not recurred for over a year. Our literature review shows that the need for surgery is not related to the location or presence of a fracture. In contrast, in the absence of osteomas, craniofacial trauma may necessitate surgery. This is the first of such findings from this literature review.

In addition to nose-blowing, coughing, sneezing, weight lifting [34], dental procedures [35] involving the use of air-driven equipment, and positive pressure ventilation [36] may also induce the development of subcutaneous emphysema [26,37,38]. A previous study reported [37] the effects of coughing, sneezing, and nose-blowing on intranasal pressure. The intranasal pressure when blowing the nose (70 mmHg) is reportedly [37] > 10 times that when sneezing and coughing (4–7 mmHg). Regardless of the cause, in orbital emphysema, the air in the paranasal sinuses is pressurized when patients blow their nose, and it leaks under the skin; therefore, observing the patient's behavior before the appearance of the peribulbar swelling may aid in diagnosis. In our review, the medial orbital wall was the most commonly fractured site, as was the case with our patient. The medial wall is

thinner than the orbital floor; thus, it may be considered more susceptible to an increase in orbital pressure.

In cases of acute onset unilateral orbital swelling, cellulitis, which may manifest as pain on eye movement and fever, is the most important differential diagnosis that should be excluded. The condition can be identified by determining whether a pressurizing action, such as nose-blowing or coughing, was performed [39]. For several patients, nose-blowing was prohibited, and conservative follow-up examination to alleviate the issue was reported [26]; however, cases that required degassing have similarly been reported [39]. With conservative treatment, the time to the disappearance of the swelling is approximately several hours to 1 month at the longest [14,25]. Symptoms apart from subcutaneous swelling, such as central retinal artery occlusion or compressive optic neuropathy may lead to vision loss or eye movement disorders [39]. In such cases, a non-conservative treatment strategy involving aspiration of the subcutaneous air should be selected.

In our case, only swelling was reported as a symptom, and while an orbital medial wall fracture was observed, no osteoma was detected; thus, we prohibited nose-blowing, and the conservative treatment was successful. This management did not differ from the approach described in the literature. If other symptoms, such as ocular disorders, had been described, we may have opted to perform a surgical procedure. To date (after more than 1 year), no case of orbital emphysema recurrence has been observed. However, surgery should still be considered to manage the medial orbital wall fracture.

In a review of cases caused by facial trauma, in 15 of the 17 cases, the time between trauma and nose-blowing was within 10 days. Therefore, it is recommended to avoid nose-blowing for approximately 2 weeks after a fracture because of an identified facial trauma.

All patients with orbital fractures should be advised that they will need to be closely monitored for visual abnormalities for the first 2 days. Furthermore, it should be explained that when visual disturbances occur, treatment of decompression may be necessary. In contrast, the frequency of the need for invasive treatment was observed to be lower in cases without facial trauma than in trauma cases in this study.

## 4. Limitations

The main limitation of this study is that it is an aggregation of case series. It is necessary to compare the prognosis of the treatment in a large number of institutions with a clearly defined treatment strategy. A limitation of our review is also the indication of surgical treatment. The incidence of ocular disorders was higher in cases [3,5,6,8,11,15–17,20,21,26,33] that required surgical treatment than in those that required conservative treatment. However, cases with ocular disorders that were improved with only conservative treatment were reported [2,4,7,9,10,12–14,18,19,22–24,27–32], among which there was a case with complete visual loss [2]. Therefore, whether surgical intervention should be considered according to the presence or severity of symptoms remains controversial.

## 5. Conclusions

We performed a literature review of 48 cases of orbital subcutaneous emphysema following nasal blowing. In cases where fractures were evident, they were often in the medial orbital wall. A strict 2-day follow-up examination of neuro-ophthalmologic findings from the onset of subcutaneous emphysema should be performed. Many trauma cases were deemed to require surgical treatment, but this was unrelated to sex or age. Non-traumatic cases require close examination for osteoma. Antibiotics were administered in 71% of cases that did not require surgical treatment, and no infections were reported. With appropriate measures, a good prognosis was reported.

**Author Contributions:** R.M., Y.M., F.M. and S.S. contributed to the conception and design of the study. R.M. and Y.M. organized the database. Y.M. and S.S. performed the statistical analysis. R.M. wrote the first draft of the manuscript. S.S., Y.M. and A.M. wrote sections of the manuscript. All authors have read and agreed to the published version of the manuscript.

**Funding:** This research received no external funding.

**Institutional Review Board Statement:** Ethical review and approval were waived for this study, as this was a single case report.

**Informed Consent Statement:** The patient provided written informed consent for the publication of this case report.

**Data Availability Statement:** Not applicable.

**Acknowledgments:** We would like to show our deep appreciation to Kajiruda (https://www.instagram.com/kajiruda_727 (accessed on 30 May 2022)) for his efforts in preparing our graphic abstract.

**Conflicts of Interest:** The authors declare no conflict of interest.

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
