# Peer review of "Subcutaneous Emphysema of the Orbit after Nose-Blowing"

_reports, doi:10.3390/reports5020021_

Round 1

Reviewer 1 Report

Thank you for presenting this interesting case and literature review lookingat an uncommon but interesting condition.

Subcutaneous emphysema of the orbit is well recognised in patients with known orbital floor fractures who blow their nose.

The importance of early followup is emphasised in your paper. The important differential to rule out in these patients is orbital cellulitis as you mention.  

Author Response

We would like to thank the reviewer for evaluating our manuscript and for his/her comment. Please note that we have sent our manuscript to an English editing company (Editage) for English proofreading. We hope that the level of English has been significantly improved in the revised manuscript.

Reviewer 2 Report

Dear Editor,

The paper by Riku Mihara entitled: “Subcutaneous emphysema of the orbit after nose-blowing.” is Review with  a clinical case.

The article is considerable scientific interest; in my opinion, it will be appreciated by readers.

The abstract summarizes and reflect the work described

The tables and figures reflect the work described in the paper.

The references are appropriate.

Major revision

-It would be appropriate to make a systematic review of the literature or a brief report, I prefer a systematic review

-the method section may be explained in detail with diagram PRISMA and statistical methods.

-An extensive English editing is needed. 

-Add a conclusion section with the data relevant for the reader

- please create a graphic abstract

Author Response

We would like to thank the reviewer for evaluating our manuscript and for his/her comment. Please note that we have sent our manuscript to an English editing company (Editage) for English proofreading. We hope that the level of English has been significantly improved in the revised manuscript.

Moreover, please note that we have added more information concerning how we performed the literature review. The added part is as follows:

“We reviewed previous articles that reported the clinical characteristics and treatment of patients with orbital emphysema. A systematic review of the literature was conducted in accordance with the PRISMA guidelines (Figure 2). We searched the PubMed database for cases diagnosed with orbital emphysema caused by nose-blowing. Our key search terms were “nose blowing emphysema” or “nose-blowing, emphysema.” We performed search for “any field,” from which 69 studies were retrieved. We also searched on Google using key words from which we retrieved six references. We also thoroughly researched the literature from references listed in Pubmed and Google online search hits. The initial search was completed at the end of September 2020. Among these 75 references, six references were conference abstract. Six non-English language articles were also removed from this review. Ten papers that did not cover the location of subcutaneous emphysema were also removed. Four papers that did not explicitly state that nose blowing was the cause were excluded. One report that did not provide detailed case information was excluded.”

In addition, we have created a PRISMA flow diagram, as per the reviewer’s insightful suggestion.

Further, we have added a “Statistical analyses” subsection, as per the reviewer’s suggestion. The added part is as follows:

“Statistical significance was determined using Fisher's exact test due to heterogeneous and small sample sizes. In all analyses, p-values of <0.05 were considered statistically significant, and a commercially available software package (JMP Pro 13, SAS Institute, Cary, NC, USA) was used for all analyses.”

Finally, we have revised the Conclusion section and created a Graphic Abstract, in accordance with the reviewer’s suggestions.

We hope that our revisions would meet the reviewer’s expectations.

Round 2

Reviewer 2 Report

Dear Editor,

The paper by Riku Mihara entitled: “Subcutaneous emphysema of the orbit after nose-blowing.” is Review with  a clinical case.

The article is considerable scientific interest; in my opinion, it will be appreciated in the present form

This manuscript is a resubmission of an earlier submission. The following is a list of the peer review reports and author responses from that submission.